METHODS AND RESOURCES

# PiVR: An affordable and versatile closed-loop platform to study unrestrained sensorimotor behavior

David Tadres[1,2,3], Matthieu Louis[1,2,4]*

1 Department of Molecular, Cellular, and Developmental Biology, University of California, Santa Barbara, Santa Barbara, California, United States of America, 2 Neuroscience Research Institute, University of California, Santa Barbara, Santa Barbara, California, United States of America, 3 Institute of Molecular Life Sciences, University of Zurich, Zurich, Switzerland, 4 Department of Physics, University of California, Santa Barbara, Santa Barbara, California, United States of America

* mlouis@ucsb.edu

**Data Availability Statement:** All data files and scripts are available from the Dryad database (accession DOI: https://doi.org/10.25349/D9ZK50).

**Funding:** This work was funded by the National Institute for Health (RO1-NS113048-01) and by the

## Abstract

Tools enabling closed-loop experiments are crucial to delineate causal relationships between the activity of genetically labeled neurons and specific behaviors. We developed the Raspberry Pi Virtual Reality (PiVR) system to conduct closed-loop optogenetic stimulation of neural functions in unrestrained animals. PiVR is an experimental platform that operates at high temporal resolution (70 Hz) with low latencies (<30 milliseconds), while being affordable (<US$500) and easy to build (<6 hours). Through extensive documentation, this tool was designed to be accessible to a wide public, from high school students to professional researchers studying systems neuroscience. We illustrate the functionality of PiVR by focusing on sensory navigation in response to gradients of chemicals (chemotaxis) and light (phototaxis). We show how *Drosophila* adult flies perform negative chemotaxis by modulating their locomotor speed to avoid locations associated with optogenetically evoked bitter taste. In *Drosophila* larvae, we use innate positive chemotaxis to compare behavior elicited by real- and virtual-odor gradients. Finally, we examine how positive phototaxis emerges in zebrafish larvae from the modulation of turning maneuvers to orient in virtual white-light gradients. Besides its application to study chemotaxis and phototaxis, PiVR is a versatile tool designed to bolster efforts to map and to functionally characterize neural circuits.

## Introduction

Since the advent of molecular tools to map and manipulate the activity of genetically targeted neurons [1,2], a major goal of systems neuroscience has been to unravel the neural computations underlying sensorimotor transformation [3,4]. Because of the probabilistic nature of behavior [5–7], probing sensorimotor functions requires stimulating an animal with reproducible patterns of sensory input that can be conditioned by the behavioral history of the animal. These conditions can be achieved by immersing animals in virtual realities [8].

University of California, Santa Barbara (startup funds). This work was also supported by the National Science Foundation under Grant No. NSF PHY-1748958, Grant No. IOS-1523125, IH Grant No. R25GM067110, and the Gordon and Betty Moore Foundation Grant No. 2919.01. The funders had no role in study design, data collection and analysis, decision to publish, or preparation of the manuscript.

**Competing interests:** The authors have declared that no competing interests exist.

**Abbreviations:** CCD, charge-coupled device; dpf, days postfertilization; Gr66a, gustatory receptor 66a; IAA, isoamyl acetate; LED, light-emitting diode; OSN, olfactory sensory neuron; PiVR, Raspberry Pi Virtual Reality; Or42a, odorant receptor 42a.

A virtual reality paradigm consists of a simulated sensory environment perceived by an animal and updated based on a readout of its behavior. Historically, virtual realities have been introduced to study optomotor behavior in tethered flies and bees [9,10]. For several decades, sophisticated computer-controlled methods have been developed to produce ever-more-realistic immersive environments. In FreemoVR, freely moving flies avoid collisions with fictive tridimensional obstacles, and zebrafish engage in social interactions with artificial peers [11]. Spatial learning has been studied in tethered flies moving on a treadmill in 2D environments filled with geometrical objects projected on a visual display [12]. The same technology has been used to record the neural activity of mice exploring a virtual space [13].

Immobilized zebrafish larvae have been studied while hunting virtual prey [14], adapting their motor responses to fictive changes in intensity of water flow speed [15] and by virtually aligning themselves with a visual moving stimulus [16]. Although virtual realities were initially engineered to study visual behavior, they have been generalized to other sensory modalities, such as touch and olfaction. In a treadmill system, navigation has been studied in mice directed by localized stimulations of their whiskers [17]. The combination of closed-loop tracking and optogenetic stimulation of genetically targeted sensory neurons has enabled a quantitative analysis of chemotaxis in freely moving *Caenorhabditis elegans* and in *Drosophila* larvae immersed in virtual-odor gradients [18,19].

Virtual reality assays aim to reproduce the natural feedback that binds behavior to sensation [8]. First, the behavior of an animal must be accurately classified in real time. In tethered flying flies, wing beat patterns have been used to deduce turning maneuvers [20]. Likewise, the movement of a tethered walking fly or a mouse can be inferred from the rotation of the spherical trackball of a treadmill [13,21]. In immobilized zebrafish larvae, recordings from the motor neuron axons have been used to infer intended forward swims and turns [16]. In freely moving *C. elegans* and *Drosophila* larvae, the posture of an animal and the position of specific body parts—the head and tail, for instance—have been tracked during motion in 2D arenas [18,19]. Variables related to the behavioral readout—most commonly, the spatial coordinates—are then mapped onto a virtual sensory landscape to update the stimulus intensity [13]. A similar methodology that enables the tracking of individual sensory organs (left and right eyes) has been recently proposed as an open-source package dedicated to zebrafish larvae [22].

The effectiveness of the virtual reality paradigm is determined by the overall temporal delay between the animal's behavior and the update of the stimulus. The shorter this delay, the more authentic the virtual reality is perceived. As a result, the methodology deployed to create efficient virtual realities with closed-loop tracking relies on advanced technology that makes behavioral setups costly and often difficult to adopt by nonspecialists. There is a scope to complement the existing collection of sophisticated assays with tools that are affordable, easily built, and accessible to most laboratories. The fly "ethoscope" proposes a hardware solution that exploits 3D printing to study the behavior of adult flies [23]. In this system, tracking and behavioral classification are implemented by a portable and low-cost computer, the Raspberry Pi. Although this system can implement a feedback loop between real-time behavioral tracking and stimulus delivery (e.g., the physical rotation of an assay to disrupt sleep), it was not conceived to create refined virtual reality environments using optogenetic stimulations.

Here, we present the Raspberry Pi Virtual Reality (PiVR) platform enabling the presentation of virtual realities to freely moving small animals. This closed-loop tracker was designed to be accessible to a wide range of researchers by keeping the construction costs low and by maintaining the basic operations simple and customizable to suit the specificities of new experiments. We benchmark the performance of PiVR by studying navigation behavior in the *Drosophila* larva. We then reveal how adult flies adapt their speed of locomotion to avoid areas associated with the activation of bitter-sensing neurons. Finally, we show that zebrafish larvae

approach a virtual-light source by modifying their turn angle in response to temporal changes in light intensity.

## Results

### PiVR permits high-performance closed-loop tracking and optogenetic stimulations

The PiVR system enables high-resolution, optogenetic, closed-loop experiments with small, freely moving animals. In its standard configuration, PiVR is composed of a behavioral arena, a camera, a Raspberry Pi microcomputer, a light-emitting diode (LED) controller, and a touch screen (Fig 1A). The platform is controlled via a user-friendly graphical interface (Fig 1B). Given that PiVR does not require the use of an external computer, the material for one setup amounts to less than US$500, with the construction costs decreasing to about US$350 when several units are built in parallel (S1 Table). As shown in S2 Table, PiVR is significantly cheaper than published alternatives that are capable of operating at equally high frame rates [11,22]. In spite of its affordability, PiVR runs a customized software (S1–S6 Figs) that automatically identifies semitransparent animals such as *Drosophila* larvae behaving in a static background (Fig 2) and that monitors movement at a frame rate sufficient to accurately track rapidly moving animals such as walking adult flies and zebrafish larvae (Figs 3 and 4).

We characterized the overall latency of PiVR in two ways. First, by measuring the following three parameters: (1) image acquisition time, (2) image processing time, and (3) the time taken for commands issued by the Raspberry Pi to be actuated by the LED hardware (S1A–S1D Fig). We find that the image processing time is the main time-consuming step. Secondly, we measured the total latency between the moment an image starts being recorded and the update of the intensity of the LED system based only on the analysis of that image. We find that the total latency is shorter than 30 milliseconds (S1E–S1G Fig). Thus, PiVR is suited to perform online tracking of small animals at a frame rate of up to 70 Hz with a lag shorter than three frames and an accuracy suitable to create virtual olfactory realities in *Drosophila* larvae [18] and virtual visual realities in walking adult flies [21].

The PiVR software implements image acquisition, object tracking, and the update of background illumination for optogenetic stimulation. It is free, fully open-source, and written in the programming language Python. At the beginning of each experiment, an autodetect algorithm separates the moving object—the animal—from the background (S3 Fig). During the rest of the experiment, the tracking algorithm operates based on a principle of local background subtraction to achieve high frame rates (S4 Fig). Besides locating the position of the animal's centroid, PiVR uses a Hungarian algorithm to tell apart the head from the tail positions (S5 Fig). For applications involving off-line tracking with a separate software [24,25], the online tracking module of PiVR can be disabled to record videos at 90 Hz in an open-loop mode.

PiVR has been designed to create virtual realities by updating the intensity of a homogeneous stimulation backlight based on the current position of a tracked animal (Fig 1C, left panel) relative to a preset landscape (Fig 1C, middle panel, S1 Movie). Virtual sensory realities are generated by optogenetically activating sensory neurons of the peripheral nervous system [1]. In the present study, we focus on applications involving CsChrimson because the red activation spectrum of this light-gated ion channel is largely invisible to *Drosophila* [26]. Depending on the light-intensity range necessary to stimulate specific neurons, PiVR features a light pad emitting stimulation light at low-to-medium (2 μW/mm$^2$) or high (22 μW/mm$^2$) intensities (S2C and S2D Fig). A key advantage of the closed-loop methodology of PiVR is that it permits the creation of virtual realities with arbitrary properties free of the physical constraints of

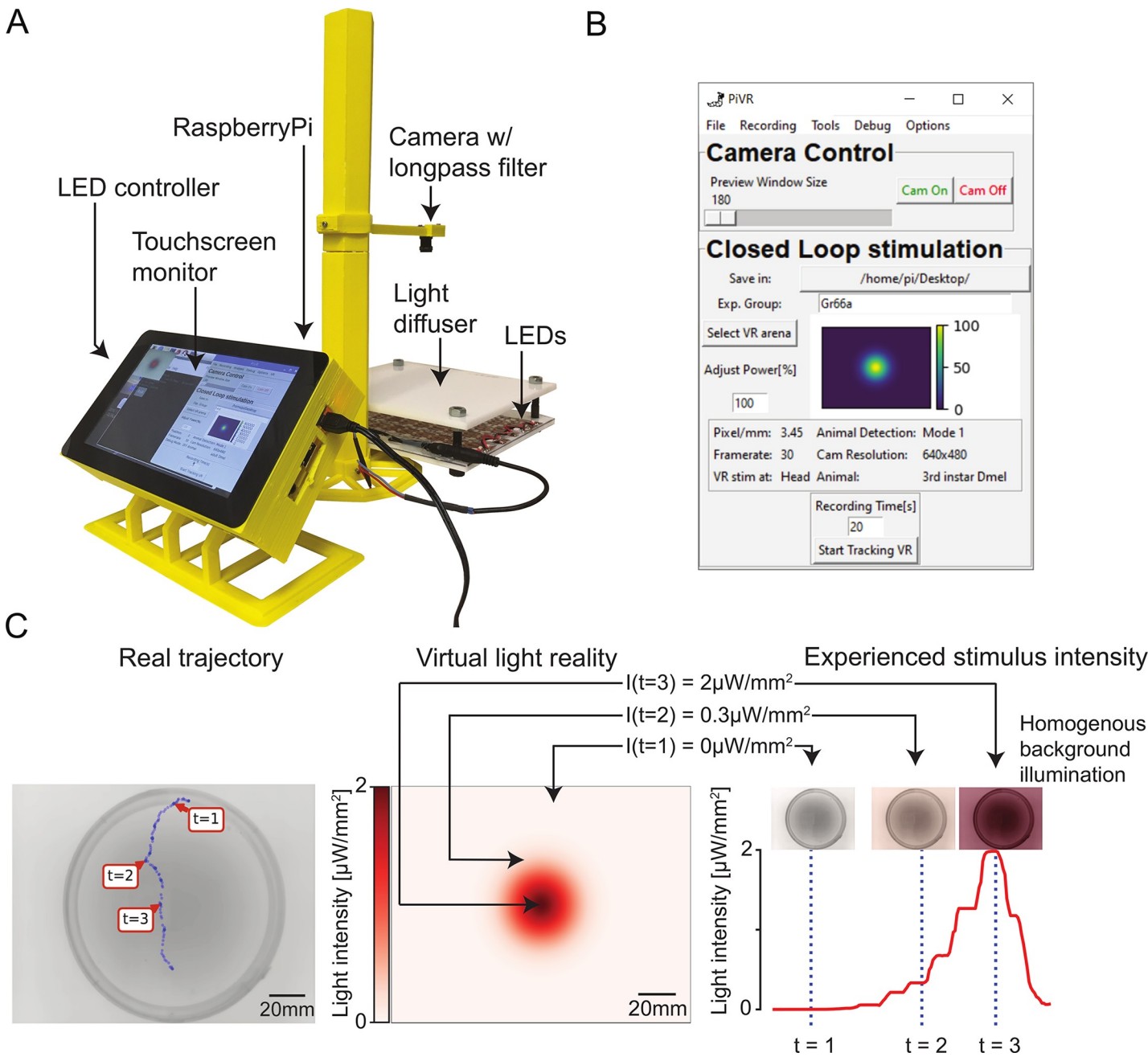

**Fig 1. Virtual realities created by PiVR.** (A) Picture of the standard PiVR setup. The animal is placed on the light diffuser and illuminated from below using infrared LEDs and recorded from above. The Raspberry Pi computer and the LED controller are attached to the touch screen, which permits the user to interface with the PiVR setup. (B) Screenshot of the GUI while running a virtual reality experiment. The GUI has been designed to be intuitive and easy to use while presenting all important experimental parameters that can be modified. (C) Virtual realities are created by updating the intensity of a homogeneous light background based on the current position of a tracked animal mapped onto a predefined landscape shown at the center. (Center) Predefined virtual gradient with a Gaussian geometry. (Left) Trajectory of an unconstrained animal moving in the physical arena. (Right) The graph indicates the time course of the light intensity experienced by the animal during the trajectory displayed in the left panel. Depending on the position of the animal in the virtual-light gradient, the LEDs are turned off (t = 1) or turned on at an intermediate (t = 2) or maximum intensity (t = 3). GUI, graphical user interface; LED, light-emitting diode; PiVR, Raspberry Pi Virtual Reality.

real stimuli. In Fig 2Fi, we illustrate the use of PiVR by immersing *Drosophila* larvae in a virtual-odor gradient that has the shape of a volcano—a geometry that challenges the sensorimotor responses of larvae in a predictable way [18].

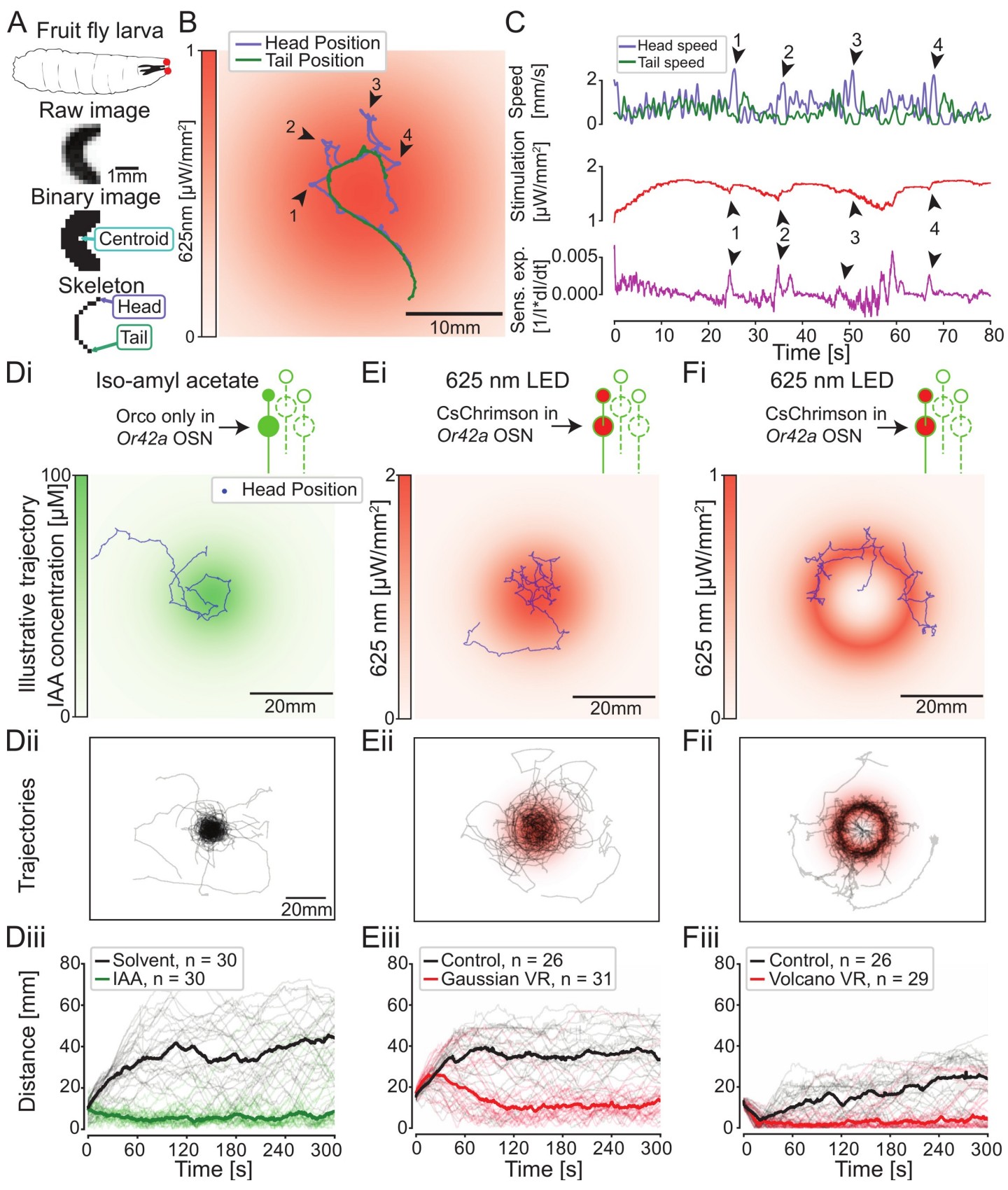

**Fig 2. Benchmarking PiVR performance by eliciting larval chemotaxis in virtual-odor gradients.** (A) *Drosophila* larva with a pair of single *Or42a*-functional OSNs (red dots). Illustration of the identification of different body parts of a moving larva by PiVR. (B) Illustrative trajectory of a larva in a Gaussian virtual-odor gradient elicited by light stimulation. Arrowheads and numbers indicate lateral head movements (casts), and the time points are congruent with the arrowheads shown in (C). Panel D shows the behavior of *Drosophila* larvae directed by *Or42a* OSNs in a gradient of IAA (green color). (E–F) Behavior of larvae expressing the light-gated ion channel CsChrimson in the *Or42a* OSNs evoking a virtual-odor gradient. In panel E, the virtual-odor gradient (red) has a geometry similar to the real-odor gradient (green) presented in (D). The "volcano" virtual-odor landscape presented in (F) highlights that the information conveyed by the *Or42a* OSN alone is sufficient for larvae to chemotax with high accuracy along the rim of the gradient. Thick lines in panels Diii, Eiii, and Fiii indicate the median distances to the source, and the light traces indicate individual trials. All data used to create this figure are available from https://doi.org/10.25349/D9ZK50. IAA, isoamyl acetate; LED, light-emitting diode; Or42a, odorant receptor 42a; OSN, olfactory sensory neuron; PiVR, Raspberry Pi Virtual Reality; Sens. exp., sensory experience; VR, virtual reality.

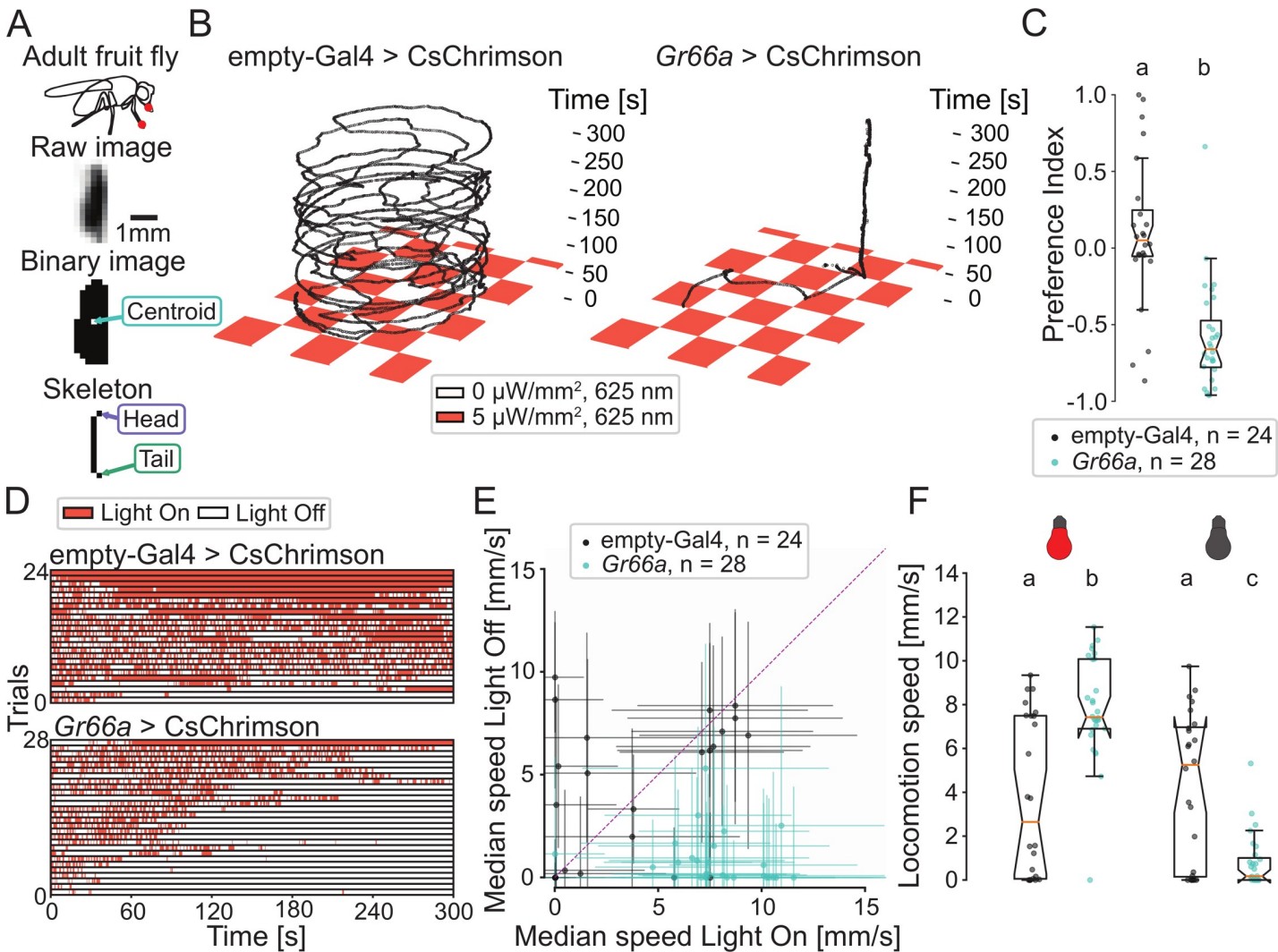

**Fig 3. Adult fruit flies avoid activation of bitter-sensing neurons by modulating their locomotion speed.** (A) Adult *Drosophila* expressing CsChrimson in *Gr66a* bitter-sensing neurons (red circles). Illustration of the identification of different body parts of a moving fly by PiVR. (B) Illustrative trajectories of flies in a virtual checkerboard pattern and the corresponding ethogram. Flies were behaving in a petri dish. (D) The ethogram reports the time spent by individual animals (rows) in the dark (white) and lit (red) squares. Panel C displays a quantification of the avoidance of virtual bitter taste through a preference index: $\text{PI} = \frac{T_{ON} - T_{OFF}}{T_{ON} + T_{OFF}}$, where T is the time spent on the ON or OFF quadrants (Mann–Whitney U test, $p < 0.001$). (E) Median locomotion speeds of individual animals as a function of the exposure to light. (F) Quantification of locomotion speeds across experimental conditions (Dunn's multiple comparisons test, different letters indicate at least $p < 0.01$). Statistical procedures are detailed in the Methods section. Statistical significances are indicated with lowercase letters. All data used to create this figure are available from https://doi.org/10.25349/D9ZK50. Gr66a, gustatory receptor 66a; PiVR, Raspberry Pi Virtual Reality.

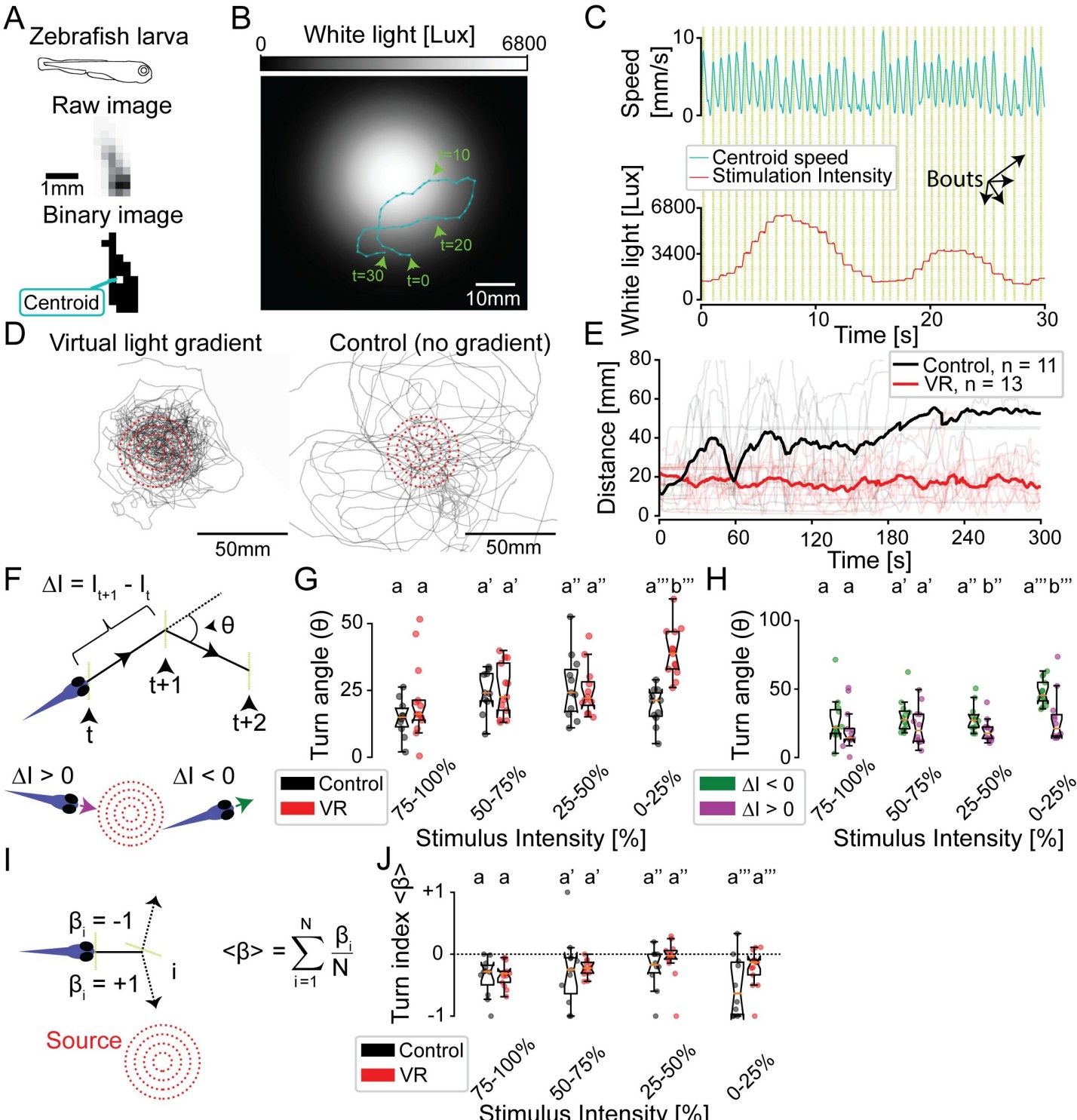

**Fig 4. Zebrafish larvae adapt their turn dynamics to stay close to a virtual-light source.** (A) Illustration of the identification of a moving zebrafish larva by PiVR. (B) Illustrative trajectory of a zebrafish larva in a virtual-light gradient having a Gaussian geometry. Panel C displays the time course of the speed and the white-light intensity during that trajectory shown in panel B. Yellow vertical lines indicate automatically detected bouts. (D) Trajectories of 13 fish tested in virtual-light gradient (left) and 11 fish in control (right). Red circles indicate 10-, 20-, 30-, and 40-mm distances to the center of the virtual-light source (see Methods). (E) Thick lines indicate the time courses of the median distances to the virtual-light source. The light lines indicate individual trials. (F) Illustration of the discretization of a trajectory segment into bouts: yellow vertical lines indicate the position at which the animal stops, reorients, and starts the next bout. Black dashed lines indicate movement of the fish. The turn angle (θ) and change in light intensity (ΔI) are calculated for every pair of consecutive bouts (see Methods). The bottom of panel F illustrates a swim bout oriented

up-gradient (purple, $\Delta I > 0$) and down-gradient (green, $\Delta I < 0$). (G) Relationship between θ and I during the previous bout (independent two-sample *t* test, different letters indicate $p < 0.001$). (H) Turn angles θ of the virtual reality condition are grouped according to negative (green) and positive (magenta) intensity experienced in the previous bout (*t* test for paired samples, different letters indicate at least $p < 0.05$). (I) The turn index (β) is calculated from the average reorientation accuracy ($β_i$) of the animal relative to the virtual-light source at the onset of each swim bout. (J) Turn index (β) as a function of stimulus intensity (Mann–Whitney U test, all groups $p > 0.05$). All reported statistical significances are Bonferroni corrected and indicated with lowercase letters. Statistical procedures are detailed in the Methods section. All data used to create this figure are available from https://doi.org/10.25349/D9ZK50. PiVR, Raspberry Pi Virtual Reality; VR, virtual reality.

The possibility to 3D print components that once required sophisticated machining has empowered the "maker movement" in our scientific community [27]. Inspired by this philosophy, PiVR is built from hardware parts that are 3D printed. Thus, the modular design of the setup can be readily adapted to accommodate the experimental needs of traditional model organisms (larvae, adult flies, and zebrafish) as well as less conventional small animals (S2 and S6 Figs). For example, we adapted PiVR to acquire movies of 10 fruit fly larvae simultaneously with an image quality sufficient to permit off-line tracking of multiple-animal tracking with the idtracker.ai software [25]. To achieve this, we modified the design of the arena to allow illumination from the side instead of bottom (S2B Fig). This adaptation of the illumination setup was necessary to enhance contrast in the appearance of individual larvae for idtracker.ai to detect idiosyncratic differences between larvae (S2Bi Fig). The versatility of PiVR was also illustrated by tracking various arthropods and a vertebrate with diverse body structures and locomotor properties (Figs 2–4, S6 Fig).

## Benchmarking PiVR performances by eliciting larval chemotaxis in virtual-odor gradients

To benchmark the performances of PiVR, we turned to the navigation behavior evoked by odor gradients (chemotaxis) in the *Drosophila* larva [28,29]. Larval chemotaxis relies on a set of well-characterized sensorimotor rules [30]. To ascend an attractive odor gradient, larvae modulate the alternation of relatively straight runs and reorientation maneuvers (turns). Stops are predominantly triggered when the larva undergoes negative changes in odor concentration during down-gradient runs. Following a stop, turning is directed toward the gradient through an active sampling process that involves lateral head movements (head casts). A key advantage of the larva as a model organism for chemotaxis is that robust orientation responses can be directed by a functionally reduced olfactory system. The odorant receptor gene 42a (*Or42a*) is expressed in a pair of bilaterally symmetric olfactory sensory neurons (OSNs) [31,32] that are sufficient to direct chemotaxis [33]. We exploited this property to compare reorientation performances elicited by real- and virtual-odor stimulations of the *Or42a* OSNs.

We started by applying the computer-vision algorithm of PiVR to track larvae with a single functional *Or42a*-expressing OSN (Fig 2A). Individual animals were introduced in a rectangular arena comprising a gradient of isoamyl acetate (IAA) at its center (Fig 2D). Once exposed to the odor gradient, *Or42a*-functional larvae quickly identified the position of the odor source, and they remained in the source's vicinity (Fig 2D and S2 Movie). The trajectories of consecutively tested larvae were analyzed by quantifying the time course of the distance between the head of the larva and the center of the odor source. The navigation of the *Or42a*-functional larvae yielded an average distance to the source significantly lower than that observed in the presence of the solvent alone (Fig 2Diii). This result is consistent with the behavior of wild-type larvae in response to attractive odors [34]. It establishes that PiVR can automatically detect and accurately track animals in real time.

Next, we tested the ability of PiVR to create virtual olfactory realities by optogenetically stimulating the larval olfactory system. In past work, robust chemotaxis was elicited in light gradients by expressing the blue-light-gated ion channel channelrhodopsin in the *Or42a*-

expressing OSN of blind larvae [18]. In these experiments, the light stimulus was delivered as a point of LED light kept focused on the larva. Using a closed-loop paradigm, the intensity of the LED light was updated at a rate of 30 Hz, based on the position of the larva's head mapped onto a landscapes predefined by the user [18]. PiVR was built on the same principle with the following modifications: (1) the spatial resolution of PiVR was reduced because the field of view of the camera captures the whole arena and not just the larva, (2) optogenetic stimulation was achieved through homogeneous background illumination instead of a light spot that must follow the larva, and (3) we favored the red-light-gated ion channel CsChrimson [26] over channelrhodopsin to minimize the innate photophobic response of larvae to the blue-light range [35].

The simplified hardware design of PiVR produced precise tracking of the head position of a larva exposed to a fictive light gradient (Fig 2B). The spatiotemporal resolution of the tracking is illustrated in Fig 2C, in which surges in head speed are associated with scanning movements of the head on a timescale shorter than 500 milliseconds [36]. Head "casts" induced transient changes in light intensity $I(t)$ [18]. These changes in stimulus intensity correspond to spikes in the relative sensory experience of the larva ($\frac{1}{I} * \frac{dI}{dt}$) (Fig 2C, arrowheads in purple trace) [30]. The tracking resolution of PiVR enabled recording periodic patterns in tail speed (Fig 2C, top green trace) that reflect consecutive cycles of acceleration/deceleration during forward peristalsis [37]. *Or42a*-functional larvae displayed strong chemotaxis in response to a point-source virtual-odor gradient (Fig 2E and S3 Movie) with a level of attraction comparable to the behavior evoked by a real-odor gradient (Fig 2D). Moreover, PiVR recapitulated the meandering trajectories along the rim of a volcano-shaped virtual-odor gradient (Fig 2F and S4 Movie) [18]. Together, the results of Fig 2 and S7 Fig validate that PiVR has the tracking accuracy and closed-loop performances necessary to elicit genuine navigation in virtual-odor gradients.

## Adult flies avoid activation of bitter-sensing neurons by modulating their locomotion speed

After having established the capability of PiVR to create virtual realities in *Drosophila* larvae, we sought to generalize the application of this tool to other small model organisms. Adult *Drosophila* are covered by a thick and opaque cuticle. Consequently, activating light-gated ion channels expressed in the sensory neurons of adult flies requires higher light intensities than in semitransparent larvae [6,26]. The background illumination system of PiVR was modified to deliver light intensities as high as 50 μW/mm$^2$ to penetrate the adult-fly cuticle [38]. Despite a 10-fold increase in locomotion speed between adult flies and larvae (peak speed of 12 mm/s and 1.6 mm/s, respectively), PiVR accurately monitored the motion of adult fruit flies for the entire duration of 5-minute trials (Fig 3A and 3B). We turn to gustation to test the ability of PiVR to evoke orientation behavior in adult flies stimulated by virtual chemical gradients.

*Drosophila* demonstrates innate strong aversion to bitter taste [39]. This behavior is mediated by a set of sensory neurons expressing the gustatory receptor gene 66a (*Gr66a*). Optogenetic activation of the *Gr66a*-expressing neurons alone is sufficient to elicit aversive responses [38]. Using the closed-loop tracking capabilities of PiVR (Fig 3A), we examined taste-driven responses of flies expressing the red-light-gated ion channel CsChrimson in their *Gr66a*-expressing neurons. Because we reasoned that navigation in response to taste might be less directed than navigation in response to airborne odors, we presented flies with a 2D landscape emulating a checkerboard (Fig 3B). In this virtual checkerboard, quadrants were associated with either virtual bitter taste (light "ON") or no taste (light "OFF"). Flies adapted their motion to avoid squares paired with virtual bitter taste (Fig 3B–3D and S5 Movie). This result generalized the field of application of PiVR to fast-moving small animals, such as walking adult flies.

To determine how flies actively avoid being exposed to bitter-tasting squares, we interrogated the spatial trajectories recorded by PiVR (Fig 3B) and correlated stimulus input with behavioral output [40]. This quantitative analysis of the relationship between stimulus dynamics and behavior highlighted that flies modulate their locomotion speed in response to excitation of their bitter-tasting neurons. When flies were located in a lit square eliciting virtual bitter taste, they moved significantly faster than when located in a dark square with no bitter taste. When flies encountered sensory relief in a dark square, they frequently stopped (Fig 3E and 3F). In summary, our results establish that PiVR is suitable to track and immerse adult flies in virtual sensory realities. Moreover, computational quantification of behavioral data produced by PiVR suggests that flies avoid bitter tastes by modulating their locomotion speed to avoid staying exposed to virtual bitter taste in the illuminated squares [41,42]. The contribution of other orientation mechanisms that integrate spatial information is left to be examined in future work. For instance, it is possible that flies implement directed turns upon their entry in a bitter-tasting quadrant (Fig 3B and S5 Movie).

## Zebrafish larvae adapt their turn amplitude to stay close to a virtual-light source

Because of its transparency and amenability to molecular genetics, the zebrafish *Danio rerio* has emerged as a tractable model system to study how sensory representations and sensorimotor transformations arise from the activity in neural ensembles in vertebrates [43,44]. Zebrafish are innately attracted by real sources of white light [45]. Here, we show that PiVR is suitable to study the organization of orientation behavior of zebrafish larvae immersed in a virtual 2D light gradient (Fig 4A and 4B). Already at the larval stage, individuals are capable of staying confined to virtual disks of white light [46]. In spite of the fact that 5-days-postfertilization (dpf) zebrafish larvae stay in constant motion, tracking with PiVR established that larvae have the navigational capabilities to stay near the peak of a virtual white-light gradient (Fig 4B and S6 Movie) by constantly returning to this position for the duration of the entire trial (Fig 4D and 4E, S8A Fig).

The elementary motor patterns (actions) underlying the behavioral repertoire of zebrafish larvae can be decomposed into stops and slow and rapid swims [47]. By analyzing the time series of the centroid speed recorded by PiVR, we observed periodic increases in swim speed (Fig 4C). These episodes correspond to bursts, or "bouts," of swim [48]. To examine the orientation strategy used by zebrafish larvae, we discretized trajectories into bouts. As described in previous work [47], each bout was found to be approximately straight (Fig 4B, cyan segments comprised by the circles). At low light intensities, significant reorientation occurred between consecutive swim bouts compared with the controls (Fig 4G).

Given that the virtual landscape produced by PiVR resulted from a temporal update of the intensity of isotropic light stimulation, we can rule out the detection of binocular differences in stimulus intensity [49]. Thus, reorientation in the virtual-light landscape of Fig 4 could only result from the detection of temporal changes in light intensity. Using the data set recorded with PiVR, we conducted a correlative analysis between the stimulus history and the reorientation maneuvers to define the visual features eliciting an increase in turning at low light intensity. More specifically, we analyzed the turn angle as a function of the light intensity for bouts associated with positive (up-gradient) and negative (down-gradient) changes in light intensity. When zebrafish larvae moved up-gradient, the turn angle was not modulated by the absolute intensity of the stimulus (Fig 4H, purple). By contrast, the turn rate increased for swim bouts oriented down-gradient at low stimulus intensity (Fig 4H, green). Therefore, we conclude that

the rate of turning of zebrafish larvae is determined by a combination of the absolute light intensity and the sign of change in stimulus intensity.

Given the ability of zebrafish larvae to efficiently return to the peak of a light gradient (Fig 4D), we asked whether zebrafish larvae can also bias their turns toward the light source. To this end, we defined the turn index ($\beta$) to quantify the percentage of turns directed toward the light gradient (Fig 4I and Methods). This metric leads us to conclude that zebrafish larvae do not bias their turns toward the source more often than the control (Fig 4J). In summary, our results demonstrate that PiVR can track zebrafish larvae at a sufficiently high spatiotemporal resolution to characterize individual swim bouts. We show that zebrafish achieve positive phototaxis by increasing the amplitude of their turns when they are moving down-gradient and experiencing low light intensities. This result is consistent with a recent in-depth study of the sensorimotor strategy controlling zebrafish phototaxis in a virtual reality paradigm [50]. We conclude that phototaxis in zebrafish is at least partially controlled by an increase in the amplitude of turns when two conditions are met: (1) the animal must detect a negative change in light intensity, and (2) the absolute light intensity must be low enough. The combination of the previous two conditions appears sufficient to generate positive phototaxis, even in the absence of turning biases toward the light gradient.

## Discussion

Anyone seeking to unravel the neural logic underlying a navigation process—whether it is the response of a cell to a morphogen gradient or the flight of a seabird guided by the Earth's magnetic field—faces the need to characterize the basic orientation strategy before speculating about its molecular and cellular underpinnings. PiVR is a versatile closed-loop experimental platform devised to create light-based virtual sensory realities by tracking the motion of unconstrained small animals subjected to a predefined model of the sensory environment (Fig 1). It was created to assist the study of orientation behavior and neural circuit functions by scholars who might not have extensive background in programming or instrumentation to customize existing tools.

### An experimental platform that is inexpensive and customizable

Prior to the commercialization of consumer-oriented 3D printers and cheap microcomputers such as the Raspberry Pi, virtual reality paradigms necessitated using custom setups that cost several thousand or even one hundred thousand dollars [11,13,18,19]. PiVR is an affordable (< US\$500) alternative that enables laboratories without advanced technical expertise to conduct high-throughput virtual reality experiments. The procedure to build PiVR is visually illustrated in S7 Movie. All construction steps are intended to be tractable by virtually any users who have access to a soldering station and a 3D printer. A detailed step-by-step protocol is available on a dedicated website (www.pivr.org).

The creation of realistic immersive virtual realities critically depends on the update frequency and the update latency of the closed-loop system. The maximal update frequency corresponds to the maximally sustained frame rate that the system can support. The update latency is the latency between an action of the tested subject and the implementation of a change in the virtual reality environment. In normal working conditions, PiVR can be used with an update frequency of up to 70 Hz with a latency below 30 milliseconds (S2 Fig). These characteristics are similar to those routinely used to test optomotor responses with visual display streaming during walking behavior in insects [21], thereby ensuring the suitability of PiVR for a wide range of applications.

Different optogenetic tools require excitation at different wavelengths ranging from blue to deep red [26,51,52]. The modularity of PiVR enables the experimenter to customize the illumination system to any wavelength range. Additionally, different animals demand distinct levels of light intensities to ensure adequate light penetration in transparent and opaque tissues. Although 5 $\mu$W/mm$^2$ of red light had been used to activate neurons located in the leg segments of adult flies with CsChrimson [38], 1 $\mu$W/mm$^2$ is enough to activate OSNs of the semitransparent *Drosophila* larva (Fig 2). In its standard version, PiVR can emit red-light intensities as high as 2 $\mu$W/mm$^2$ and white-light intensities up to 6,800 Lux—a range sufficient for most applications in transparent animals (Figs 2 and 4). For animals with an opaque cuticle, we devised a higher-power version of the backlight illumination system that delivers intensities up to 22 $\mu$W/mm$^2$ (525 nm) and 50 $\mu$W/mm$^2$ (625 nm) (Fig 3 and S1D Fig). Given that an illumination of 50 $\mu$W/mm$^2$ is approaching the LED eye safety limits (International Electrotechnical Commission: 62471), it is unlikely that experimenters will want to exceed this range for common applications in the lab.

## Exploring orientation behavior through virtual reality paradigms

By capitalizing on the latest development of molecular genetics and bioengineering [1], virtual sensory stimuli can be created by expressing optogenetic tools in targeted neurons of the peripheral nervous system of an animal. In the present study, PiVR was used to immerse *Drosophila* in virtual chemosensory gradients (Figs 2 and 3). In a first application, we stimulated one OSN of *Drosophila* larvae with CsChrimson. The attractive search behavior elicited by a single source of a real odor (Fig 2Di) was reproduced in an exponential light gradient (Fig 2Ei). To reveal the precision with which larvae orient their turns toward the gradient, larvae were tested in a virtual-odor landscape with a volcano shape (Fig 2Fi).

Adult flies move approximately 10 times faster than larvae. We established the ability of PiVR to immerse freely moving flies in a virtual bitter-taste gradient (Fig 3B). Unlike the attractive responses elicited by appetitive virtual odors in the larva (positive chemotaxis), bitter taste produced strong aversive behavior (S5 Movie). A correlative analysis of data recorded with PiVR revealed that the aversive behavior of adult flies is at least partly conditioned by a modulation of the animal's locomotor speed: random search is enhanced upon detection of (virtual) bitter, whereas locomotion is drastically reduced upon sensory relief. As illustrated in Fig 3, PiVR offers a framework to explore the existence of other mechanisms contributing to the orientation of flies experiencing taste gradients. This approach adds to earlier studies of the effects of optogenetic stimulation of the bitter taste system on spatial navigation [38] and feeding behavior [53].

Zebrafish move through discrete swim bouts. The loop time of PiVR was sufficiently short to rise to the tracking challenge posed by the discrete nature of fish locomotion (Fig 4B). Genuine phototactic behavior was elicited in zebrafish larvae for several minutes based on pure temporal changes in light intensity devoid of binocular differences and a panoramic component. By synthetically recreating naturalistic landscapes [45], the behavior recorded by PiVR in Gaussian light gradients (Fig 4) complements previous studies featuring the use of discrete light disks [46] and local asymmetric stimulations [54]. A correlative analysis of the sensory input and the behavioral output corroborated the idea that positive phototaxis in fish can emerge from a modulation of the turn rate by the detected changes in light intensity (Fig 4G and 4H) without necessarily involving a turning bias toward the light gradient (Fig 4J). This orientation strategy shares similarities with the nondirectional increase in locomotor activity ("dark photokinesis") that follows a sudden loss of illumination [55,56]. Taken together, our results establish that PiVR is suitable to conduct a detailed analysis of the sensorimotor rules

directing attractive and aversive orientation behavior in small animals with distinct body plans and locomotor properties.

## Outlook

Although this study focused on sensory navigation, PiVR is equally suited to investigate neural circuits [4] through optogenetic manipulations. To determine the connectivity and function of circuit elements, one typically performs acute functional manipulations during behavior. PiVR permits the time-dependent or behavior-dependent presentation of light stimuli to produce controlled gain of functions. The use of multiple setups in parallel is ideal to increase the throughput of behavioral screens—a budget of US$2,000 is sufficient to build more than five setups. If the experimenter wishes to define custom stimulation rules—triggering a light flash whenever an animal stops moving, for instance—this rule can be readily implemented by PiVR on single animals. For patterns of light stimulation that do not depend on the behavior of an animal—stimulations with a regular series of brief light pulses, for instance—groups of animals can be recorded at the same time. In its standard configuration (Fig 1A), the resolution of videos recorded with PiVR is sufficient to achieve individual tracking of group behavior through off-line analysis with specialized algorithms such as idtracker.ai [25] (S2Bi Fig). Even for small animals such as flies, videos of surprisingly good quality can be recorded by outfitting the charge-coupled device (CCD) camera of PiVR with appropriate optics (S8 Movie).

Until recently, systems neuroscientists had to design and build their own setup to examine the function of neural circuits, or they had to adapt existing systems that were often expensive and complex. Fortunately, our field has benefited from the publication of a series of customizable tools to design and conduct behavioral analysis. The characteristics of the most representative tools are reviewed in S2 Table. The ethoscope is a cost-efficient solution based on the use of a Raspberry Pi computer [23], but it was not designed to create virtual realities. Several other packages can be used to track animals in real time on an external computer platform. For instance, Bonsai is an open-source visual programming framework for the acquisition and online processing of data streams to facilitate the prototyping of integrated behavioral experiments [57]. FreemoVR can produce impressive tridimensional virtual visual realities [11]. Stytra is a powerful open-source software package designed to carry out behavioral experiments specifically in zebrafish with real-time tracking capabilities [22]. As illustrated in comparisons of S2 Table, PiVR complements these tools by proposing a versatile solution to carry out closed-loop tracking with low-latency performance. One limitation of PiVR is that it produces purely homogenous temporal changes in light intensity without any spatial component. Because of its low production costs, the simplicity of its hardware design, and detailed documentation, PiVR can be readily assembled by any laboratory or group of high school students having access to a 3D printer. For these reasons, PiVR represents a tool of choice to make light-based virtual reality experiments accessible to experimentalists who might not be technically inclined.

We anticipate that the performance (S1 Fig) and the resolution of PiVR (Methods) will keep improving in the future. Historically, a new and faster version of the Raspberry Pi computer has been released every 2 years. In the near future, the image processing time of PiVR might decrease to just a few milliseconds, pushing the frequency to well above 70 Hz. Following the parallel development of transgenic techniques in nontraditional genetic model systems, it should be possible to capitalize on the use of optogenetic tools in virtually any species. Although PiVR was developed for animals measuring no more than a few centimeters, it should be easily scalable to accommodate experiments with larger animals such as mice and rats. Together with FlyPi [58] and the ethoscope [23], PiVR represents a low-barrier

technology that should empower many labs to characterize new behavioral phenotypes and study neural circuit functions with minimal investment in time and research funds.

## Methods

### Hardware design

We designed all 3D-printed parts using 3D Builder (Microsoft Corporation). An Ultimaker 3 (Ultimaker, Geldermalsen, the Netherlands) with 0.8-mm print cores was used to print all parts. We used 2.85-mm PLA (B01EKFVAEU) as building material and 2.85-mm PVA (HY-PVA-300-NAT) as support material. The STL files were converted to gCode using Ultimaker's Cura software (https://ultimaker.com/en/products/ultimaker-cura-software). The printed circuit boards (PCBs) were designed using Fritzing (http://fritzing.org/home/) and printed by AISLER BV (Lemiers, the Netherlands).

### Hardware parts

Raspberry Pi components were bought from Newark element14 (Chicago, United States) and Adafruit Industries (New York, US). The 850-nm long-pass filter was bought from Edmond Optics (Barrington, US). Other electronics components were obtained from Mouser Electronics (Mansfield, US), Digi-Key Electronics (Thief River Falls, US), and Amazon (Seattle, US). Hardware was obtained from McMaster (Elmhurst, US). A complete bill of materials is available in S1 Table. Updates will be available on www.pivr.org and https://gitlab.com/louislab/pivr_publication.

### Building and using PiVR

Detailed instructions on how to build a PiVR and how to use it can be found in S1 HTML. Any updates will be available on www.pivr.org.

### Data analysis and statistics

All data and scripts have been deposited as a data package on Dryad: https://doi.org/10.25349/D9ZK50 [59]. Data analysis was performed using custom written analysis codes, which are bundled with the data package. In addition, the data analysis scripts are available from https://gitlab.com/LouisLab/PiVR.

### Latency measurements of PiVR

To estimate the time it takes between an animal performing an action and PiVR presenting the appropriate light stimulus, the following elements were taken into account: (1) image acquisition time, (2) image processing and VR calculation latency, and (3) software-to-hardware latency. To measure image acquisition time (S1B Fig), the camera was allowed to set optimal exposure at each light intensity before measuring the shutter speed. To measure image processing and VR calculation latency (S1Ci Fig), time was measured using the non-real-time Python time.time() function, which is documented to have uncertainty in the order of a few microseconds. To confirm these measurements, we also recorded the timestamps given by the real-time graphical processing unit (GPU) as reported in S1Cii Fig. Together, these measurements show that although image processing time has a median around 8–10 milliseconds, there are a few frames for which the image processing time takes longer than 20 milliseconds, which, at high frame rates, leads to frames being skipped or dropped (arrows in S1Ci Fig and S1Cii Fig). Finally, to measure the software-to-hardware latency, we measured how long it takes to turn the general-purpose input/output (GPIO) pins ON and OFF during an

experiment (S1D Fig). The pins are connected to a transistor with rise and fall times in the order of microseconds (https://eu.mouser.com/datasheet/2/308/FQP30N06L-1306227.pdf). LEDs tend to have a latency in the order of 100 ns. To confirm these measurements, we used PiVR to measure total time between movement in the field of view of the camera and the LED being turned on. The standard tracking software was modified to operate according to the following rules: (1) turn the LED "ON" at the 50th frame (and multiples thereof) and (2) compare pixel intensity of a fixed region of interest of the image to the previous image. If the value is above a threshold, count the frame as "flash." Turn the LED OFF again. (3) If the LED was turned OFF in the previous frame, turn it ON again. The complete code can be found on the PiVR GitLab repository on branch "LED_flash_test." This paradigm allowed us to quantify the maximal latency between animal behavior on the arena and the update of the LED. We found that the LED was always turned ON while the next image was collected as the time to detection depended on the observed location in the frame (S1E and S1G Fig). Therefore, the maximal latency is <30 milliseconds.

## Video recording performance of PiVR

When PiVR is used as a video recorder, the resolution (px) limits the maximal frame rate (fps): at 640 × 480 px, the frame rate can be set up to 90 fps, at 1,296 × 972 up to 42 fps, at 1,920 × 1,080 up to 30 fps, and at 2,592 × 1,944 a frame rate up to 15 fps can be used. PiVR is compatible with a wide variety of M12 lenses, allowing for high-quality video recordings, depending on the experimental needs (S8 Movie).

## Fruit fly larval experiments

For the experiments using fruit fly larvae (Fig 2), animals were raised on standard cornmeal medium at 22°C on a 12-hour day/night cycle. Third instar larvae were placed in 15% sucrose for 20–120 minutes prior to the experiment. Experiments were conducted on 2% agarose (Genesee, 20–102). In the experiments described in Fig 2, the arena was a 100-mm-diameter petri dish (Fisher Scientific, FB0875712).

For the experiments featuring a real-odor gradient, IAA (Sigma Aldrich, 306967–100 ML) was diluted in paraffin oil (Sigma Aldrich, 18512-1L) to produce a 1 M solution. A single source of the odor dilution was tested in larvae with the following genotype: *w;Or42a-Gal4; UAS-Orco,Orco$^{-/-}$*. The control consisted of solvent (paraffin oil) devoid of odor. For the virtual reality experiments, the following genotype was used: *w;Or42a-Gal4,UAS-CsChrimson; UAS-Orco,Orco$^{-/-}$*. In Fig 2, the same genotype was used in controls, but the tests were conducted without any light stimulations. Larvae expressing CsChrimson were grown in complete darkness in 0.5 M all-*trans* retinal (R2500, MilliporeSigma, MO, USA).

## Adult fruit fly experiments

Male flies with the *Gr66a*-Gal4 transgene (Bloomington stock number: 57670) [60] were crossed to virgin females carrying *20xUAS-CsChrimson-mVenus* [26] integrated into the *attP40* landing site. The flies were grown in complete darkness on standard cornmeal medium with 0.5M all-*trans* retinal at 25°C. Female flies between 1 and 7 days after eclosion were selected after putting the vial on ice for a few seconds. The experiment was conducted in a 100-mm-diameter petri dish (Fisher Scientific, FB0875712) under a white-light condition.

## Zebrafish larva experiments

In the experiments shown in Fig 4, we used AB *casper* [61] as parents. Only pigmented larvae were used for the experiments. The larvae were reared at 28.5˚C and a 14:10 light cycle. The experiments were run at 26˚C. The arena was a 100-mm petri dish (Fisherbrand, 08-757-12PK). All ambient light was blocked. The maximum white-light intensity provided by PiVR was measured to be approximately 6,800 Lux (Extech Instruments Light Meter 401025).

## Data analysis and statistical procedures

All data analysis was performed using Python. The scripts used to create the plots shown in the figures (including all the data necessary to recreate the plots) can be found at (https://doi.org/10.25349/D9ZK50). Generally, data sets were tested for normal distribution (Lilliefors test) and for homogeneity of variance (Levene's test) [62]. Depending on the result, the parametric *t* test or the nonparametric Mann–Whitney U rank-sum test was used. To compare multiple groups, either Bonferroni correction was applied after comparing multiple groups, or Dunn's test was applied [62]. Below, information about the analysis and the applied statistical tests throughout the manuscript are separately addressed for each figure. To estimate peak movement speed of different animals, the median of the 90th percentile of maximum speed per experiment was calculated.

**Data analysis of Fig 2.** To calculate movement speed, the *x* and *y* coordinates were first filtered using a triangular rolling filter with a window size equal to the frame rate (30 fps) divided by the high bound on the speed of the animal (1 mm/s) times the pixel-per-millimeter value of the experiment. Depending on the exact distance between camera and the arena (pixel-per-millimeter value), the window size of the filter was typically 0.3 seconds. Speed was calculated using Euclidian distance. The time series of the speed was smoothened using a triangular rolling filter with a window size of 1 second. To calculate the sensory experience, the stimulus intensity time course was filtered using a boxcar rolling filter with window size 1 second (Fig 2C). To calculate the distance to source for the IAA gradient, the source location was manually defined using the PiVR software. In the Gaussian virtual-odor gradient, the coordinates with the maximum intensity value was defined as the source. In the volcano-shaped virtual gradient, the circle with the highest values was defined as the nearest source to the animal (Fig 2F). At 4 minutes into the experiment, the distance to source between the experimental and control condition was compared. As the data were not normally distributed (Lilliefors test), Mann–Whitney U test was used (S7A–S7C Fig).

**Data analysis of Fig 3.** Each experiment lasted 5 minutes. The preference index for each animal was calculated by subtracting the time spent by an animal in the squares without light from the time spent in the squares with light (squares eliciting virtual bitter taste). This subtraction was then divided by the total experimental time (Fig 3C). Mann–Whitney U test was used to compare preference between genotypes because the distribution of the preference indices was not normally distributed (Lilliefors test). Speed was calculated as described in Fig 2: first, the *x* and *y* coordinates were smoothened using a triangular rolling filter with a window size equal to the frame rate (30) divided by the approximate walking speed of flies (approximately 5 mm/s) times the pixel-per-millimeter ratio of the experiment, which was usually 0.06–0.09 seconds. Locomotion speed was calculated by using the Euclidian distance. The time series of the speed itself was filtered using a triangular rolling filter with a window size equal to the frame rate (30 fps). To test for statistical difference in speed between genotypes and the light on and off condition, Dunn's multiple comparison test was implemented through the scikit-posthocs library of Python.

**Data analysis of Fig 4.** Each experiment lasted 5 minutes. Each trial started with the animal facing the virtual-light source. To focus the analysis on animals that demonstrated significant movement during the experiment, the final data set was based on trials fulfilling the following criteria: (1) an animal had to move at least 20 mm over the course of the experiment, (2) only movements above 1 mm per frame were recorded (due to camera/detection noise, the centroid in resting animals can move), and (3) the animal must have been tracked for at least 3 out of 5 minutes.

Locomotion speed was calculated by first smoothening the *x* and *y* centroid coordinates with a half-triangular rolling filter with the window size of 1 second. The speed was calculated using Euclidian distance and filtered again using a 0.3-second triangular rolling filter. Swim bouts were identified from the time course of the movement speed by using the "find_peaks" function of the scipy library [63] with the following parameters: a minimum speed of 2.5 mm/s and a minimum time between two consecutive peaks (bouts) of five frames (0.165 seconds) (Fig 4C). The same filter was applied (half-triangular rolling filter with window size of 1 second) to calculate the distance to source. Distance to the maximum value of the virtual reality landscape was calculated. To calculate the distance to source of the controls, the same virtual reality was assigned relative to the animal starting position, and the distance to this simulated landscape was calculated (Fig 4E). In S8A Fig, the distance to source was compared across experimental conditions at 4 minutes into the experiment. We used Mann–Whitney U test to compare the distance to source because the variance between samples could not be assumed to be equal (Levene's test) (S8A Fig). Bouts (see above) were then used to discretize the trajectory. The reorientation angle θ was calculated for each pair of consecutive bouts by comparing the animal's coordinates 0.66 seconds (approximate duration of a bout) before and after the local peak in locomotor speed. To filter out occasional (<10%) misdetection of the animal's reflection once it was located near the wall of the petri dish, we only considered bouts with a reorientation angle θ smaller than 135˚ (Fig 4F). The light intensity experienced by the animal was used to bin the reorientation angles into four groups, shown in Fig 4G and 4H. To compare the control versus experimental condition of the four light intensity bins, Student *t* test was used after checking for normality (Lilliefors test) and for equality of variance (Levene's test). Bonferroni correction was used to correct for multiple comparisons (Fig 4G).

To compare reorientation (turn) angles θ associated with positive and negative visual sensory experiences, the change in light intensity experienced during the previous bouts (ΔI) was used to split the experimental data in two groups: ΔI > 0 (up-gradient) and ΔI < 0 (down-gradient). The data were binned according to stimulus intensity as in Fig 4G. To compare turn angle in the two groups, Student *t* test for paired samples was used after checking for normality (Lilliefors test) and for equality of variance (Levene's test). Bonferroni correction was used to adjust for multiple comparisons (Fig 4E).

The turn index β was calculated based on the angular difference α between the heading of the animal at a given bout and the bearing with respect to the white-light source. This angular difference α indicates whether the animal approaches the source (angles near 0˚) or swims away from it (angles near 180˚). We defined 90˚ as the boundary between swimming toward and swimming away from the source. The turn index β was then defined by counting the number of bouts toward the source and by subtracting from it the number of bouts away from the source normalized by the total number of bouts. A turn index was calculated for each trial. We compared each group using the Mann–Whitney U test because the data were not normally distributed (Lilliefors test). Bonferroni correction was used to adjust for multiple testing (Fig 4J). To bin the reorientation (turn) angle θ according to distance to source, the distance was calculated for each bout and binned in four groups according to S8C Fig. We compared each group

using Mann–Whitney U test because the data were not always normally distributed (Lilliefors test). Bonferroni correction was used to adjust for multiple comparisons.

## Supporting information

**S1 Fig. Timing performance of PiVR.** (A) Illustration of the three parameters measured to estimate overall loop time and latency. (B) Dependence of image acquisition time on the infrared background illumination strength. (Ci) To measure image processing and VR computation time, the non-real-time python 3.5 time.time() function is used. At 50, 60, and 70 fps some frames (6/5,598, 25/7,198 and 16/8,398, respectively) take longer than the period of the frame rate, which leads to dropped frames. (Cii) To confirm these measurements, we also recorded timestamps of the images assigned by the real-time clock of the GPU. (D) To estimate the software-to-hardware latency during a full update cycle of the tracker, we measured the time between the GPIO pin being instructed to turn ON and the GPIO pin reporting being turned ON. (E) Cameras with a rolling shutter take images by reading out lines of pixels from top to bottom (E, left). This can be illustrated in a simple $x/y$ plot (E, right). To estimate maximum latency between the animal position and the update of the LED intensity, we used an LED flash paradigm while PiVR was tracking a dummy object at 70 Hz. Our latency results depend on the ROI associated with the location of the dummy object. When PiVR tracks an ROI located at the top of the frame (Fi, left), it detects the LED only two frames later (Fii, left). By contrast, when PiVR tracks an ROI located at the bottom of the frame (Fi, right), it detects the LED in the next possible frame (Fii, right). If PiVR tracks an ROI in the center of the image (Fi, center), it either detects the LED during the next frame or two frames later. We conclude that the LED is being turned ON while the next image is being formed (1.34 milliseconds). For a full LED ON–LED OFF–LED ON sequence, we find that there are three frames between the LED being turned ON when PiVR tracks the top of the image (Fiii, left), whereas it takes two frames when PiVR tracks the bottom of the image (Fiii, right). Because the time between two light flashes contains the frame during which the LED is turned OFF, the image acquisition and processing time corresponds to one or two frames. This is summarized in the timeline of panel G. Taken together, this shows that the time necessary to detect the movement of an animal and to update the LED intensity takes a maximum of two frames plus the time necessary to take the picture, which amounts to a maximum of 30 milliseconds at a frame rate of 70 Hz. All data used to create these plots are available from https://doi.org/10.25349/D9ZK50. GPIO, general-purpose input/output; GPU, graphical processing unit; LED, light-emitting diode; PiVR, Raspberry Pi Virtual Reality; ROI, region of interest; VR, virtual reality. (TIF)

**S2 Fig. Modularity of PiVR (hardware).** (A) PiVR is highly modular. The standard version (shown in Fig 1A) can easily be adapted to allow for side (or top) illumination using the same LED controller system. If high-light intensities are needed, a high-power LED controller can be used in combination with high-power LEDs. Panel B shows an example of a side illuminator. Side illumination increases contrast on the surface of the animal. (Bi) This side illuminator was used to collect videos with sufficient detail for idtracker.ai [25] to track 10 fruit fly larvae while retaining their identity. (C) The standard PiVR illuminator consists of at least two different 12-V LED strips: one for background illumination (850 nm) and another color (here, 625 nm) to stimulate the optogenetic tool of choice. (D) The high-power stimulation arena uses an LED controller that keeps current to high-power LEDs constant and can drive a maximum of 12 high-power LEDs. (Ci and Di) The intensity of the stimulation light was measured using a Photodiode (Thorlabs Inc. S130VC). Each pixel is 2 cm$^2$. The black circles indicate petri dishes

used as behavioral arenas. LED, light-emitting diode; PiVR, Raspberry Pi Virtual Reality.
(TIF)

**S3 Fig. Flow diagram of the automatic animal detection and background reconstruction.**
(A) After placing the animal and pressing "start tracking" on the graphical user interface, the software will grab the first image. All images are immediately filtered using a Gaussian kernel with a sigma depending on the size of the animal (user defined) to reduce camera noise. (B) Then a second image is taken. The mean of the images taken so far is being calculated. (C) The current image (in this example, the second image of the experiment) is then subtracted from the mean image shown in panel B. (D) The histogram of the subtracted image shows most gray scale values to be 0. For the region where the animal has moved since the first frame, the pixel values are negative (magenta). For the region that the animal has left since the first frame, the pixel values are positive (green). (E) The threshold value is calculated based on the histogram: it is the mean of the image subtracted by 4 (optimal value defined by trial and error). (F) The threshold value is used to binarize the subtracted image shown in panel C. If there is no or more than one blob with a minimal area (defined by user in animal parameters file), the loop restarts at step (B). (G) If there is exactly one blob, an area around the blob (defined by user in animal parameters file) is defined as the current region of interest. (H) The region of interest is the area where movement has been detected. The algorithm will now restrict the search for the animal to this region. (I) The histogram of this small area of the first image (A) shows that the few pixel defining the animal are distinct from the background. (J) To find the optimal local threshold for binarizing the image, the threshold is adjusted if more than one blob is detected (top, green arrow). As soon as only one blob with the characteristics of the animal (defined by user in animal parameters file) has been detected, the local threshold value is set, and the shape of the identified animal in the first frame is saved (bottom). (K) Using the local threshold, each new image is binarized and then subtracted from the first image as shown in panel (J, bottom). If the identical blob that was detected in panel J (bottom) is found in any of the new subtracted binary images (cyan arrow), the animal is considered as having left its original position, and the algorithm continues. (L) The region occupied by the animal is then copied from the latest image and (M) pasted into the first image. (N) The resulting image does not contain the animal and will be used as the background image for the tracking algorithm (S4 Fig). All data used to create these plots are available from https://doi.org/10.25349/D9ZK50.
(TIF)

**S4 Fig. Flow diagram of the animal tracking algorithm.** (A) At the start of the experiment, the ROI is defined during animal detection (S3G Fig). During the experiment, the current ROI is defined using the previous frame. The ROI of the current image (C) is then subtracted from the ROI of the background (B). The fact that the tracking algorithm only considers a subsample of the image is central to the temporal performances (short processing time) of PiVR. (D) In the resulting image, the animal clearly stands out relative to the background. (E) The histogram of the image indicates that whereas the background consists mostly of values around 0, the animal has pixel intensity values that are negative. (F) The threshold is defined as being three standard deviations away from the mean (G). This threshold is used to binarize the subtracted ROI. The largest blob with animal characteristics (defined by animal parameters) is defined to be the animal. (H) The image of the detected animal is saved, and the next ROI is designated (defined by animal parameters). All data used to create these plots are available from https://doi.org/10.25349/D9ZK50. PiVR, Raspberry Pi Virtual Reality; ROI, region of interest.
(TIF)

**S5 Fig. Head/tail classification using a Hungarian algorithm.** (A) During tracking, head/tail classification starts with the binarized image (S4G Fig). (B) The binary image is used to calculate the morphological skeleton, which in turn is used to identify the two endpoints, one of which must be the head and the other the tail. (C) The Euclidian distance between the tail position in the previous frame and each of the endpoints is calculated. If the tail was not defined in the previous frame, the centroid position is used instead. (D) Whichever endpoint has less distance is defined as the tail (here $v$). The other endpoint is defined as the head.
(TIF)

**S6 Fig. Capability of PiVR to track a wide variety of small animals.** PiVR is able to detect, track, and assign head and tail positions to a variety of invertebrate species with different body plans: (A) kelp fly, (B) jumping spider, (C) firefly, and (D) pill bug. PiVR, Raspberry Pi Virtual Reality.
(TIF)

**S7 Fig. Distance to real- and virtual-odor sources in larval chemotaxis to real- and virtual-odor gradients.** (A) Distance to real-odor source (isoamyl acetate, $n = 30$) and the solvent (paraffin oil, $n = 30$), (B) between the Gaussian-shaped virtual-odor reality ($n = 31$) and the control ($n = 26$), and (C) the distance to the local maximum (rim of the volcano, $n = 29$) and the control ($n = 26$). Time point is 4 minutes into the experiment (Mann–Whitney U test, $p < 0.001$). All data used to create these plots are available from https://doi.org/10.25349/D9ZK50.
(TIF)

**S8 Fig. Zebrafish larvae in virtual-light source.** (A) Distance to virtual-light source of control (black) and experimental condition (red) at 4 minutes into the experiment. (B) Relationship between turn angle θ and distance to the virtual-light source (Mann–Whitney U test, different letters indicate $p < 0.01$, $n = 11$ and 13). All reported $p$-values are Bonferroni corrected. All data used to create these plots are available from https://doi.org/10.25349/D9ZK50.
(TIF)

**S1 Movie. Illustration of homogenous illumination creating a virtual checkerboard reality.** The behavior of a freely moving fly is shown in the petri dish (left), in the virtual checkerboard recorded by PiVR (middle). The corresponding time course of the homogenous illumination intensity is shown in the (right) panel. PiVR, Raspberry Pi Virtual Reality.
(MP4)

**S2 Movie. Sample trajectory of a *Drosophila* larva expressing *Orco* only in the *Or42a* olfactory sensory neuron behaving in a quasistatic isoamyl acetate gradient.** The odor gradient was reconstructed for visualization and an estimation of the experienced odor intensity as described before [33] (see also Methods section). Or42a, odorant receptor 42a.
(MP4)

**S3 Movie. Illustrative trajectory of a *Drosophila* larva expressing the optogenetic tool CsChrimson in the *Or42a* olfactory sensory neuron behaving in a Gaussian-shaped virtual-odor reality.** Or42a, odorant receptor 42a.
(MP4)

**S4 Movie. Illustrative trajectory of a *Drosophila* larva expressing the optogenetic tool CsChrimson in the Or42a olfactory sensory neuron in a volcano-shaped virtual-odor reality.** Or42a, odorant receptor 42a.
(MP4)

**S5 Movie. Illustrative trajectory of an adult *Drosophila* expressing the optogenetic tool CsChrimson in the *Gr66a* bitter-sensing neurons in a checkerboard-shaped virtual gustatory reality.** Gr66a, gustatory receptor 66a.
(MP4)

**S6 Movie. Illustrative trajectory of a zebrafish larva in a virtual visual reality.**
(MP4)

**S7 Movie. Time-lapse video illustrating the production and assembly of a PiVR setup ending with the start of an experiment.** For a detailed step-by-step protocol, please visit www. pivr.org. PiVR, Raspberry Pi Virtual Reality.
(MP4)

**S8 Movie. Video sequences of an adult fly in a dish recorded with PiVR outfitted with three different lenses.** Comparison of the image quality between the standard 3.6-mm, F1.8 lens that comes with the Raspberry Pi camera; a 6-mm, F1.8 lens (US$25, PT-0618MP); and a higher-magnification 16-mm, F1.8 lens (US$21, PT-1618MP). PiVR, Raspberry Pi Virtual Reality.
(MP4)

**S1 Table. Bill of materials for the standard version of PiVR.** PiVR, Raspberry Pi Virtual Reality.
(CSV)

**S2 Table. Comparative analysis of prevalent tracking systems used for behavioral analysis.** (1) Optogenetic activation in closed-loop experiments. FreemoVR and Stytra are designed to present complex 3D and 2D visual stimuli, respectively. (2) Maximal frequency of the closed-loop stimulus. (3) Maximal time between an action of the tracked animal and the update of the hardware presenting the virtual reality. (4) Most tracking algorithms monitor the position of the centroid. Some tools will automatically detect other features of the tracked animal. (5) Some tracking algorithms are designed to specifically identify animals with a stereotypic shape, size, and movement. Other tracking algorithms are more flexible. (6) Assessed based on published information. (7) Bonsai was used in the optoPAD system, which uses optogenetics [53]. The information presented in this comparative table relies on the following publications: FlyPi [58]; Ethoscope [23]; FreemoVR [11]; Bonsai [57]; PiVR, present manuscript. PiVR, Pi Raspberry Virtual Reality.
(PDF)

**S1 HTML. Copy of the www.pivr.org website, which includes information on how to build a standard PiVR setup.** PiVR, Raspberry Pi Virtual Reality.
(ZIP)

## Acknowledgments

We thank Ellie Heckscher, Primoz Ravbar, and Andrew Straw for comments on the manuscript. We are grateful to Ajinkya Deogade for work performed during the initial phase of the project (development of a do-it-yourself open-loop tracker based on FlyPi), for creating a preliminary draft of 3D-printed parts, for measuring the white-light intensities used in Fig 4, and for discussions. We thank Tanya Tabachnik for technical advice about the assay construction. We are grateful to Stella Glasauer for collecting the kelp fly tracked in S6A Fig, for taking the picture of the spider in S6B Fig, for designing the PiVR logo, and for comments on the manuscript. We are in debt to Tyler Sizemore for collecting the firefly and pill bug tracked in S6 Fig.

We thank Igor Siwanowicz for providing the pictures of the firefly and pill bug of S6 Fig. We are grateful to Minoru Koyama and Jared Rouchard for rearing and providing the fish used in Fig 4. The development and optimization of PiVR greatly benefited from feedback received during the Drosophila Neurobiology: Genes, Circuits & Behavior Summer School in 2017, 2018, and 2019 at the Cold Spring Harbor Laboratory.

## Author Contributions

**Conceptualization:** David Tadres, Matthieu Louis.

**Data curation:** David Tadres.

**Formal analysis:** David Tadres.

**Funding acquisition:** Matthieu Louis.

**Investigation:** David Tadres, Matthieu Louis.

**Methodology:** David Tadres.

**Project administration:** Matthieu Louis.

**Resources:** David Tadres.

**Software:** David Tadres.

**Supervision:** Matthieu Louis.

**Validation:** David Tadres.

**Visualization:** David Tadres.

**Writing – original draft:** David Tadres, Matthieu Louis.

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
