## [Editor Report · Decision Letter 0]

19 Dec 2019

Dear Matthieu, 

Thank you for submitting your manuscript entitled "PiVR: an affordable and versatile closed-loop platform to study unrestrained sensorimotor behavior" for consideration as a Methods and Resources by PLOS Biology.

Your manuscript has now been evaluated by the PLOS Biology editorial staff as well as by an academic editor with relevant expertise and I am writing to let you know that we would like to send your submission out for external peer review.

Please re-submit your manuscript within two working days, i.e. by Dec 23 2019 11:59PM.

***Please be aware that, due to the voluntary nature of our reviewers and academic editors, manuscripts may be subject to delays due to their limited availability during the holiday season. Please also note that the journal office will be closed entirely 21st- 29th December inclusive, and 1st January 2020. Thank you for your patience.***

Kind regards,

Ines

--

Ines Alvarez-Garcia, PhD

Senior Editor

PLOS Biology

Carlyle House, Carlyle Road

Cambridge, CB4 3DN

+44 1223–442810

---

## [Decision Letter · Decision Letter 1]

31 Jan 2020

Dear Matthieu,

Thank you very much for submitting your manuscript "PiVR: an affordable and versatile closed-loop platform to study unrestrained sensorimotor behavior" for consideration as a Methods and Resources at PLOS Biology. Thank you also for your patience as we completed our editorial process, and please accept my apologies for the delay in providing you with our decision. Your manuscript has been evaluated by the PLOS Biology editors, an Academic Editor with relevant expertise, and by three independent reviewers.

As you will see, the reviewers feel that the system you have developed is novel and useful for the scientific community, however they also raise several issues that would need to be addressed before we can consider your manuscript further for publication. You should make a comprehensive comparison with other methods already available (such as Bonsai, Stytra or FlyPI) and state clearly the advantages and limitations of PiVR. In addition, you should streamline the text and follow the reviewers’ suggestions to improve the structure of the manuscript.

In light of the reviews (attached below), we are pleased to offer you the opportunity to address the [comments/remaining points] from the reviewers in a revised version that we anticipate should not take you very long. We will then assess your revised manuscript and your response to the reviewers' comments and we may consult the reviewers again.

We expect to receive your revised manuscript within 1 month.

**IMPORTANT - SUBMITTING YOUR REVISION**

*Resubmission Checklist*

*Published Peer Review*

*PLOS Data Policy*

Sincerely,

Ines

--

Ines Alvarez-Garcia, PhD

Senior Editor

PLOS Biology

Carlyle House, Carlyle Road

Cambridge, CB4 3DN

+44 1223–442810

Reviewers’ comments

Rev. 1:

The present study by Tadres and Louis presents a useful and affordable open-source virtual reality setup based on the Raspberry Pi system. The authors present solid data using virtual chemotaxis and phototaxis experiments, confirming that their system works accurately for such purposes. I find this work valuable to the community, especially for the effort to design systems that can be implemented at low cost as well as for teaching platforms.

I have a few comments that I believe would improve the presentation and message of this manuscript and one concern regarding the proposed speed of the system.

First, I have my doubts that the overall close-loop reaction time is correct. The shutter speeds and the image processing time were measured. However, due to the limitations of the bus, there is always a lag between image acquisition and the start of image processing. I have no experience with the Raspberry Pie system, but I have encountered these limitations in top of the line computing systems. Thus, I would test this directly. My suggestion is to use their system at maximal acquisition speed, image short LED flash, and use their computing platform to detect this flash and turn the same LED on for a second time. By looking at the number of frames between the two flashes, the authors could test the speed of the entire system. Although the sampling rate will be limited by the acquisition speed of the camera, the authors will be able to prove if their proposed 20ms processing time is accurate. This is important since the authors correctly claim that "the shorter the delay, the more authentic the virtual reality is perceived." I don't see issues with the experiments presented, but new users will be warned in case the system is slower than the speeds required for their purposes.

My second suggestion is regarding the current structure. The main claim of this paper is to have a VR system that is affordable and reliable; not the experimental results used to benchmark the system. Thus, I would remove the experimental sections from the discussion (Defining that nature of taste-driven responses in adult flies; Exploring the ability of zebrafish [larvae] to orient with minimal graded visual inputs; Exploring the ability of the fruit fly….) and merged them in a minimal form to the experimental results.

Finally, although I truly appreciate and are convinced that the system is useful for particular experiments, it won't be for all. There are limitations on the FOV, camera resolution, speed, etc., that make it indispensable for many labs to engineering their systems. Thus, I would recommend adding this perspective to the discussion. I believe that this will be helpful for labs that thanks to the affordable design will start doing these kinds of VR experiments, but don't have the experience to judge all details.

Minor:

I would disagree with the first sentence: "Behavior emerges from the conversion of sensory input into motor output". Organisms are not automata, e.g.; there is a whole world of internal states that modulate behavior. Please rephrase.

I would also make a stronger point on why VR systems are important (line 41). You don't need a VR system to stimulate an animal repeatedly. VR systems allow combining the stimulus to the behavior, enabling a precise exploration of response dynamics to sensory stimuli dependent on particular behavioral conditions.

- The paper sometimes uses colloquial terms, e.g., "crack the neuronal code," etc. I would change that language.

- Hz and frames per second are used, I would stick to Hz.

- Reference, e.g., page 213: Schulze, Gomez-Marin (17) -> Schulze et al. (17). Also line 307.

- Get rid of popular in line 220 and 256. Model organisms already imply that they are used widely.

- Line 600 - 608, it is not clear that the (Ci-iii) refers to the supplemental figure.

- Line 195, "field of the view", remove the.

Rev. 2:

In this manuscript, Tadres & Louis present PiVR, a novel hardware designed to perform closed-loop light-based stimulations on small animals. The authors present data based on optogenetic activation of genetically targeted neurons in Drosophila larvae and imagos; they also present a proof of principle using visible light on zebrafish larvae.

PiVR is certainly an interesting device and I can see it becoming popular in the field of Drosophila neuroscience, especially among people interested in developing new paradigms of learning.

The device has some clear strengths and some limitations. The strengths are discussed appropriately and fairly. The limitations not so much and it may be useful to have a more rounded discussion of both so that readers can immediately recognise whether this tool is appropriate for their uses.

Amongst the strengths I would count:

1. The documentation is outstanding.

2. The machine is relatively inexpensive.

3. The basic usage of the machine seems to be easy to implement

Amongst the weaknesses:

1. Some of the proposed usages of PiVR are suboptimal. It is stressed several times in the manuscript that PiVR can be used to acquire videos "offline", for them to be tracked by a different software (such as tracker.ai). I doubt this setup can compete, in terms of resolution, speed and even cost with the simpler solution of having an industrial CCD connected to an existing computer.

2. While, in principle, the R in VR stands for any kind of Reality, in fact it is commonly associated to complex visual representations such as projections of objects, patterns, scenarios. PiVR is limited to providing different intensity of lights and therefore its main use-case is going to be optogenetics.

3. It is not clear to me why a reader should prefer PiVR over other already existing alternatives and it may be useful to stress out, perhaps in a table, pros and cons of PiVR vs the "competition". FlyPI , BONSAI, FreeMO-VR are all alternative products with functionalities that overlap the ones of PiVR - the manuscript would benefit from a more straight comparison with them.

I do not have specific comments on the manuscript, besides the fact that I found it perhaps too long and repetitive. I think the strength of this tool is that it does one thing and it does nicely but all this gets somehow lost in a manuscript so discursive and repetitive. The discussion, in particular, is not well focused.

The figures are well presented and clear but perhaps not very focused. figures 2 and 3 read more like a demonstration that "optogenetics works". It would have been more useful to focus on PiVR versatility and show several different use cases rather than only those two very basic ones. Somehow the focus of the figures is optogenetics, not PiVR.

I also had troubles understanding part of figure 4 (F-J) and almost all of figure 5. Perhaps the concepts of slow vs fast dynamic VR can be explained in greater detail with a cartoon and examples of why and how an experimenter may want to use slow or fast dynamic VR could be provided.

Rev. 3:

This paper describes an open, low-cost platform to perform closed-loop behavioral experiments. The intention is to make such experiments more accessible or scalable by removing barriers of cost and development. The utility of the system is thoroughly demonstrated through example experiments in larval and adult Drosophila and zebrafish, which are analyzed to reveal new biological insights.

The resources described in the paper are well designed and described and should be straightforward to apply to many current experimental questions. However, there are several existing platforms with overlapping aims and many of them are not cited or discussed in this paper. Some of these allow much more experimental flexibility than the generation of virtual environments through modulation of a one-dimensional signal that is used in the current paper. Therefore, it is not clear that the method fulfills the criteria required for a resource in PLOS Biology, in that it would enable experiments that were not possible using existing methods. In particular, it would be helpful to cite the following two resources and discuss them in comparison with the PIVR system:

1) Stytra

http://www.portugueslab.com/pages/resources.html

Stytra is a python-based package, which can be used for animal tracking and closed-loop presentation of stimuli. Implementations with low-cost hardware are possible. Štih V*, Petrucco L*, Kist AM, Portugues R (2019)

Stytra: An open-source, integrated system for stimulation, tracking and closed-loop behavioral experiments. PLOS Computational Biology, doi:10.1371/journal.pcbi.1006699

2) Bonsai.

http://www.kampff-lab.org/bonsai

Bonsai is a software framework that is widely used for designing closed-loop behavior experiments, and can be used in conjunction with low cost hardware.

Bonsai: An event-based framework for processing and controlling data streams. Lopes G, Bonacchi N, Frazão J, Neto JP, Atallah BV, Soares S, Moreira L, Matias S, Itskov PM, Correia PA, Medina RE, Calcaterra L, Dreosti E, Paton JJ, Kampff AR. Frontiers in Neuroinformatics. 2015; 9:7.

Some minor issues:

1) In the introductory discussion on virtual reality in neuroscience, it could be worth citing the productive use of virtual reality in work in zebrafish, in the context of motor adaptation, prey capture and social behavior.

2) The closed-loop experiments based on optogenetic stimulation of the gustatory system in flies have technical similarities with experiments in the following paper, which might be appropriate to cite:

Moreira, J.-M., Itskov, P. M., Goldschmidt, D., Baltazar, C., Steck, K., Tastekin, I., et al. (2019). optoPAD, a closed-loop optogenetics system to study the circuit basis of feeding behaviors. eLife, 8. http://doi.org/10.7554/eLife.43924

3) The terminology used in the zebrafish swimming experiments in not aligned to common usage in the literature, and could be confusing. Individual bursts of movement of the larvae are typically called 'bouts' , and the word 'scoots' is used to refer to a particular type of bout, where the tail oscillations are mostly confined to the caudal region, and which propels the larva forward at a slow speed, without major reorientation or head yaw (also often called 'Slow swims'). Routine turns, by contrast, start with a larger tail movement that reorients the larva, which can be followed by a propulsive phase. Therefore what are called 'scoots' in this paper may encompass both 'scoots' and 'turns' in the terminology of other studies.

4) In the discussion of phototaxis in zebrafish, there are some other papers that perhaps are relevant to cite, because they either describe virtual reality based assays for phototaxis, or discuss the choice of turns vs scoots and the direction of turns. Ahrens, M. B., Huang, K. H., Narayan, S., Mensh, B. D., & Engert, F. (2013). Frontiers in Neural Circuits, 7, 104.

Fernandes, A. M., Fero, K., Arrenberg, A. B., Bergeron, S. A., Driever, W., & Burgess, H. A. (2012). Current Biology, 22(21), 2042-2047. Horstick, E. J., Bayleyen, Y., Sinclair, J. L., & Burgess, H. A. (2017). BMC Biology, 1-16.

---

## [Editor Report · Decision Letter 2]

20 Mar 2020

Dear Matthieu,

Thank you for submitting your revised Methods and Resources entitled "PiVR: an affordable and versatile closed-loop platform to study unrestrained sensorimotor behavior" for publication in PLOS Biology. I have now discussed the revision with the team of editors and obtained advice from the original Academic Editor. 

We're delighted to let you know that we're now editorially satisfied with your manuscript. However before we can formally accept your paper and consider it "in press", we also need to ensure that your article conforms to our guidelines. A member of our team will be in touch shortly with a set of requests. As we can't proceed until these requirements are met, your swift response will help prevent delays to publication. Please also make sure to address the data and other policy-related requests noted at the end of this email.

*Copyediting*

*Published Peer Review History*

*Early Version*

*Submitting Your Revision*

Best wishes,

Ines

--

Ines Alvarez-Garcia, PhD

Senior Editor

PLOS Biology

Carlyle House, Carlyle Road

Cambridge, CB4 3DN

+44 1223–442810

DATA POLICY:

You may be aware of the PLOS Data Policy, which requires that all data be made available without restriction: http://journals.plos.org/plosbiology/s/data-availability. I can see that you have deposited your data in Driad (DOI: https://doi.org/10.25349/D9ZK50), but the link doesn't seem to be active and I can't check if the data we need before the manuscript enters production is available. Please either activate it or follow the instructions stated below.

Note that we do not require all raw data (for more information, please also see this editorial: http://dx.doi.org/10.1371/journal.pbio.1001797). Rather, we ask that all individual quantitative observations that underlie the data summarized in the figures and results of your paper be made available in one of the following forms:

Fig. 3C, F; Fig. 4G, H, J; Fig. S1B, C, D; Fig. S3E, D, I, J; Fig. S4E, F; Fig. S7A, B, C and Fig. S8A, B

Please ensure that your Data Statement in the submission system accurately describes WHERE YOUR DATA CAN BE FOUND.

---

## [Editor Report · Decision Letter 3]

9 Jun 2020

Dear Dr Louis,

On behalf of my colleagues and the Academic Editor, Tom Baden, I am pleased to inform you that we will be delighted to publish your Methods and Resources in PLOS Biology. 

Early Version

PRESS 

Kind regards,

Alice Musson

Publishing Editor, 

PLOS Biology

on behalf of

Ines Alvarez-Garcia,

Senior Editor

PLOS Biology